# Developing a Risk Management Process for Infrastructure Projects Using IDEF0

Hui-Ping Tserng [1,*], I-Cheng Cho [1], Chun-Hung Chen [2] and Yu-Fan Liu [3]

1   Department of Civil Engineering, National Taiwan University, No. 1, Sec. 4, Roosevelt Road, Taipei 10617, Taiwan; uscedula@gmail.com
2   Second District Project Office, Department of Rapid Transit Systems, Taipei City Government, No. 7, Lane 48, Sec. 2, Zhongshan N. Rd, Taipei 10537, Taiwan; chchen@dorts.gov.taipei
3   CTCI Smart Engineering Corporation, No. 16, Lane 270, Sec. 3, Bei Shen Road, New Taipei City 11155, Taiwan; smallvanx@gmail.com
*   Correspondence: hptserng@gmail.com

**Abstract:** The Mass Rapid Transit (MRT) project is a massive, large-scale construction venture with a complex interface. In order to reduce the risk of disasters and industrial accidents in the project and to save costs, a simple and flexible risk management system is necessary for projects such as MRT. A set of risk management processes was identified through a literature review and data collection, and the Integration Definition for Function Modeling (IDEF0) process was used for logical analysis. The IDEF0 diagram clearly depicts the items to be delivered at each interface, and risk is reduced by facilitating the flow of data on various risk items. The results of this research will be applied to other practical projects, with special emphasis on the project planning and design stages. Future work will verify whether the implementation of the proposed risk management process does indeed effectively reduce risks in the completed project.

**Keywords:** risk management; risk process; project management; IDEF0; risk system implementation

## 1. Introduction

Uncertainty in a project is a source of risk [1], and the complicated and changeable environment of the construction industry is associated with high uncertainty and thus high risks. Moreover, projects must be completed within a limited time frame [2]. Public construction projects are large-scale with complex environments and long durations, so the uncertainty is much higher and more difficult to control than other types of projects [3,4]. Many uncertain circumstances are encountered in the project implementation process, and engineers and project managers are often forced to make decisions in emergency situations [5,6]. Risk management plays an important role in contract management, and thus, managers must have the knowledge to adequately carry out risk management [7,8].

Risk management has become a very important part of project management. Its scope of application has expanded beyond the traditional practice and is no longer limited to the construction phase [9,10]. In other words, extra effort should be put into the management of risk in public construction projects [11], and the complete management process must include risk identification, risk analysis, and the disposition of each risk item to minimize disasters and losses [12,13].

Traditional construction management focuses only on construction progress, project quality, and expenses, including cost and time. The effects of these three items depend on the overall risk management in each phase of the project cycle, including the planning, design, and construction phases [14,15]. If all risk events are properly controlled, then the construction project will run smoothly and meet quality requirements. The project can also be completed within the estimated cost and time without additional expenses that cause budget overflows.

Therefore, in recent years, risk management has been gradually receiving more attention in the civil engineering field. It is increasingly being applied to different types of construction projects to prevent predictable risks and reduce losses [9]. Public construction projects have a huge impact on the national economy, and the occurrence of disasters during construction results in incalculable social costs and life and property losses [16,17]. Moreover, the quality of risk management has a dramatic impact on the operational quality of the facility upon completion of the project [18].

## 2. Problem Statement

In general, the life cycle of a public construction project can be divided into the following phases. The first phase is the "Feasibility Assessment" [19]. After this stage of assessment, if there are implementation benefits, the project proceeds to the next stage, namely, the "Planning Stage". The third stage is the "Design Stage", which is usually divided into two parts: "Basic Design" and "Detailed Design", in which basic principles and detailed designs, respectively, are established for the project. The fourth and most important stage of the project is the "Construction Stage". This stage also has a direct impact on the success of the project. Finally, the last stage is the "Operational Phase", in which the community can enjoy the results of the project.

There are various risks involved in all stages of the construction project, from the feasibility assessment to the operational phase. Although great efforts are made to resolve the risks at their emerging stages, residual or unresolved risks are shifted or added to the next phase of the construction life cycle. Currently, there are no explicit rules concerning the handover of risks from one stage to the next. However, each stage contains some form of transferred risk. For example, during the preparation of procurement contract documents for the design stage, design requirements are specified. These requirements are the risk management results obtained from the planning stage. These risk items are handed over to the design stage. Then, the supervision unit controls particular risks in the construction stage. Therefore, risk management should cover the whole life cycle of construction. Disasters that evolve from risks in construction projects may occur at any stage of the life cycle. Thus, risk management is a very important topic in this industry. Avoiding the repeated occurrence of similar risks that can cause disasters at different stages of the life cycle process and preventing the incorrect transmission of such risks are research subjects that continue to expand.

## 3. Research Objectives

Risk management is being increasingly studied. The development of risk management procedures utilizes past knowledge and experience [20,21], and using the "risk-based approach" is an important success factor in project management [22]. Few studies have performed in-depth analyses of the risk management process. In addition, the methods used in risk management differ between companies, and it is difficult to preserve data because the duration of each project is very long. These factors make the flow of information between contractors and projects ineffective, even if there is adequate historical information on risk. Different contractors may make the same mistakes and need to increase costs to resolve disasters caused by recurring risks. Therefore, it is necessary to systematically study historical risk data and develop a risk management procedure [12,20].

Currently, there are no specific requirements or formats for the approach to risk transmission between stages. Therefore, developing specifications or uniform standards for risk transmission is important. The aim of this study is to construct a comprehensive risk management process for public construction projects with a common language of communication. For this purpose, all risk management information should be shared through a common platform so that all parties have access to all risk management information, allowing them to make the necessary decisions in the shortest possible time at any stage of the construction project. By achieving this goal, future project participants can more successfully manage risks, and the incidence of engineering disasters can be reduced.

## 4. Research Background and Literature Review

The literature review reveals that a large portion of risk management involves study of the probability of the occurrence of risks, or the problems that affect the disasters when the risks occur. It is rarely discussed whether some treatment can be done in the previous stage to prevent the occurrence of risks. Whether some important matters are ignored due to incomplete message transmission in the preceding and following stages, which triggers the occurrence of risk events. The traditional risk management focuses only on the construction phase [9,10]. The risk management on construction stage focuses only on construction progress, project cost and expenses. It ignores that the effects of these three items depend on the overall risk management at all stages of the project cycle, including planning, design, and construction stages [14,15]. The risk management primarily focuses on the effectiveness of the process for a single project. Traditional management methods mainly focus on risk stages or strategy execution [23,24]. But In this study found that risk management must be continuously applied in a feedback loop, and risk control and monitoring are performed via data management systems [25,26]. One of the reasons that risk management fails is the absence of risk management procedures or their improper application [23,27,28]. Currently, risk management does not focus on the relationship between stakeholders and needs to be repeated and constantly monitored. It is necessary to include the risk management process as a topic of discussion.

In this study, the IDEF0 methodology was used with a specific focus on the relationship between stakeholders [29] and on the identification of the input and output information products at each stage to prevent disasters caused by the asymmetry of information between stages [30]. The IDEF0 analysis method is the most clear and effective approach to defining the products to be delivered by different contractors at different stages of the lifecycle.

## 5. Methodology

The process followed in this research is shown in the flowchart in Figure 1, and the details are as follows:

1. Introduction

The reason that public construction projects need risk management is explained.

2. Problem statement

Efficient communication between all stakeholders is necessary at different stages to manage risk from the perspective of the project lifecycle.

3. Research objectives and background

Decision-makers can make correct judgments based on information exchanged on a common platform. The probability of disasters can be reduced by preventing different contractors from making the same mistakes at different stages.

4. Literature review and methodology

The IDEF0 analysis method is the most clear and effective approach to defining the products to be delivered by different contractors at different stages. An expert grading method is implemented using occurrence probability and impact magnitude coupled with a risk matrix to define risk level.

5. System implementation

A common management method is used, and the data transfer process occurs through a database on a common platform for different contractors at different stages.

6. Case study

The implementation of an actual construction project is used as a practical case study. The construction period of this project is seven years.

7. Conclusions

After this research is complete, the accuracy of the evaluation can be improved.

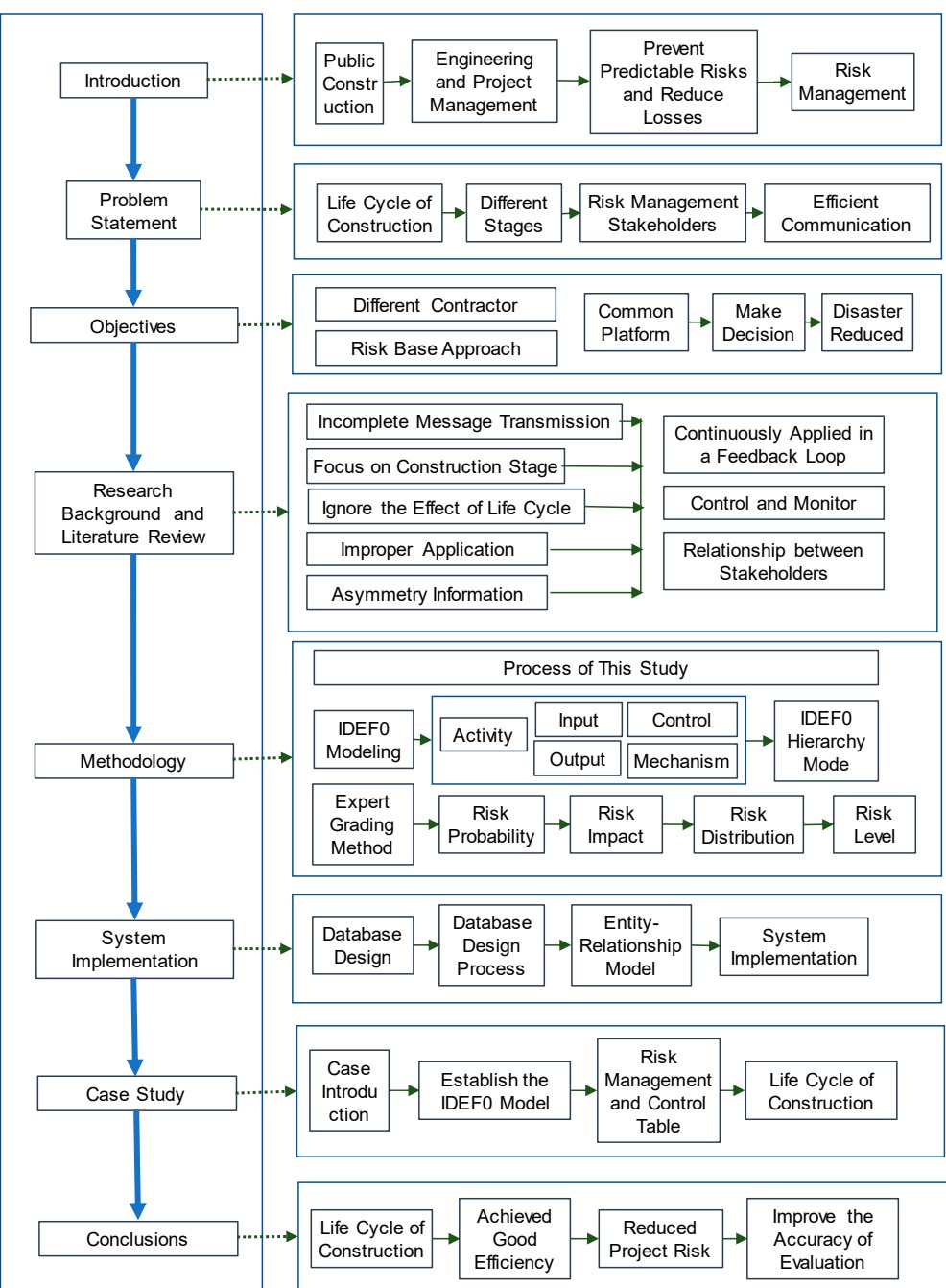

**Figure 1.** Research flowchart.

The Integration Definition for Function Modeling (IDEF0) was used in this study to illustrate the risk management workflow at a glance and to analyze the different units at various stages of the project life cycle. After the identification of risk items in each stage, an expert grading method was used to determine the risk level using the occurrence probability and impact magnitude coupled with a risk matrix to define the risk level. Failure Mode and Effect Analysis (FMEA) is recognized as one of the most valuable techniques in reliability and risk management [31,32]. FMEA is a structured technique that can help to identify all failure modes within a system, assess their impact, and plan corrective actions, and it has been widely used in the construction industry [33]. Fuzzy logic and fuzzy analytical hierarchy process (AHP) are used to address the limitations of traditional FMEA in the construction industry [34]. The following is a detailed description of the modular IDEF0 analysis and expert grading method.

*IDEF0 Modeling for the Risk Management Process*

In 1977, Ross and Schoman proposed structural analysis and design techniques, and in 1978, the U.S. military adopted the approach to support the Incident Cause Analysis Method (ICAM). This resulted in the creation of the IDEF0 methodology, which consists of a series of methods to support the modeling of the business process or inter-industry demand patterns [35]. The IDEF0 methodology includes a total of 16 methods, from IDEF0 to IDEF14. Each method has its own application field, and they provide mutual support to each other. These methods enable a holistic analysis, design, and diagnostic solution in an enterprise or an organization, and they can act as a tool for communication between different work teams [36]. IDEF0 can be used to identify the important programs of the project. The whole system is broken down into different work activities from the top to the bottom, and the result shows the required information and resources for each activity, including hardware, software tools, and human resources.

For an existing system in operation, IDEF0 can be used to analyze and record the actual operation of each activity in the system. For a completely new system, IDEF0 can first define the requirements of the system and design and implement the system according to these requirements. In this study, the IDEF0 model was used to develop a new system.

The IDEF0 model consists of three different information types—graphic, text, and vocabulary, which are cross-referenced to each other. Each IDEF0 graphic contains 3–6 boxes in a ramp-like arrangement. The boxes and arrows form an ICOM (input, control, output, mechanism) map, which includes input, control, output, and mechanism items, as shown in Figure 2. Each ICOM map can be divided into several sub-maps, which, to further clarify the items and the structure of the map, include structured text that describes the features, processes, and links between boxes. The vocabulary is used to define the keywords in the graphics.

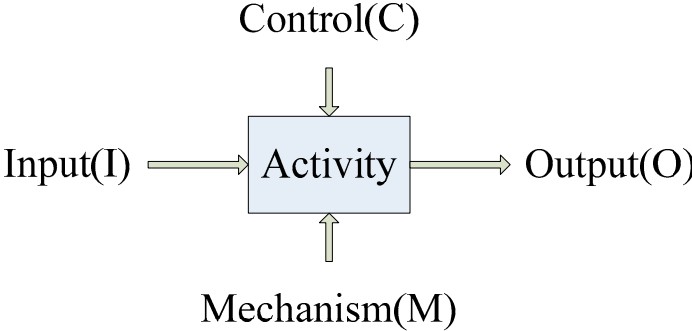

**Figure 2.** IDEF0 structural diagram (modified from Integration Definition for Function Modeling (IDEF0), 1993).

The IDEF0 method can systematically describe a complicated manufacturing system by decomposing it from top to bottom. The IDEF0 graphic is simple, clear, and readable. Therefore, it is very easily understood by management and manufacturing personnel and can assist the system analyst in explaining the current system and the proposed ideal system to the relevant management personnel, as shown in Figure 3. In this project risk management, 15 charts using IDEF0 are used to show the design, construction and operation phases of the life cycle.

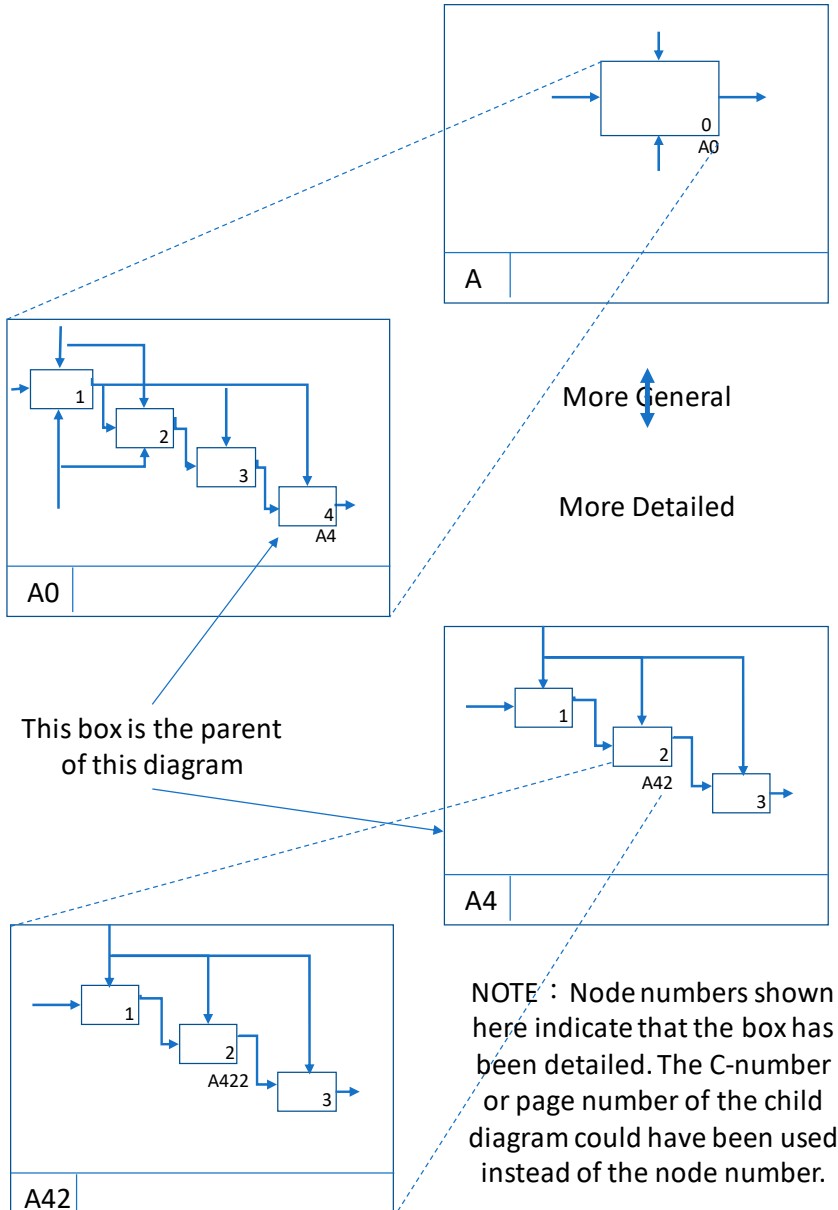

**Figure 3.** IDEF0 model hierarchy (modified from Integration Definition for Function Modeling (IDEF0), 1993) grading method.

Once the risk management process has been established, risk assessment is conducted. The main method adopted in our risk assessment process is the expert grading method. The process involves a comprehensive assessment of risk factors, responses and measures, and the levels of impact. The identified risks can be used to design an assessment checklist, which can be used by the evaluator team to review and score the probability, level, and impact of the particular risk event. Then, risk elimination and minimization measures are recommended. The risk matrix combines the probability of the risk occurrence and the level of risk. Different responses and measures are prescribed for different risks, depending on their risk level. All data are recorded in a risk management and control summary table.

The evaluation team consists of members of the risk assessment team (established by the assessment design unit) and external expert consultants; each evaluator has a different specialization and over 10 years of professional experience. The evaluation team can conduct a risk event assessment based on the scope of the evaluation specifically established for the project. The preliminary assessment results proposed by the evaluation

team and expert consultants are then reviewed and recompiled during the assessment team meeting to ensure consistency in the risk assessment results.

The risk probability is preliminarily determined based on the risk probability levels proposed by the International Tunneling and Underground Space Association (ITA). The probabilities are presented in five categories, which are, in ascending order, 'Very unlikely', 'Unlikely', 'Occasional', 'Likely', and 'Very likely', which are denoted by the indicators P1–P5, respectively. Table 1 details the risk probability levels and categories. The probability of each classification is based on suggestions modified from ITA, 2004.

**Table 1.** The risk probability levels and categories (modified from ITA, 2004).

| Probability Classification | | |
|---|---|---|
| Classification | Indicators | Probability |
| Very likely | P5 | >0.3 |
| Likely | P4 | 0.03–0.3 |
| Occasional | P3 | 0.003–0.03 |
| Unlikely | P2 | 0.0003–0.003 |
| Very unlikely | P1 | <0.0003 |

The risk impact is the impact severity of a particular risk. The impact levels are 'minor', 'limited', 'severe', 'very severe', and 'catastrophic', which are denoted by the indicators G1–G5, respectively. There are three factors to consider when determining the risk impact: (1) injury or death during the project or project failure, (2) adverse impact on the project schedule, and (3) the ratio of the business' financial loss to the total project cost. Table 2 details the risk impact levels and categories.

**Table 2.** The risk impact levels and categories (modified from ITA, 2004).

| Consequence Classification Table | | | | | |
|---|---|---|---|---|---|
| Impact Levels | | | | | |
| Risk Impact | Catastrophic G5 | Very Severe G4 | Severe G3 | Limited G2 | Minor G1 |
| Injury or death during project or project failure | F > 10 | $1 < F \leq 10$ SI > 10 | 1F $1 < SI \leq 10$ | 1SI $1 < MI \leq 10$ | 1MI |
| Adverse impact on project schedule | >24 months | 6–24 months | 2–6 months | 1/2–2 months | <1/2 months |
| The ratio of the business' financial loss to the total project cost | >33% | 3.3–33% | 0.33–3.3% | 0.03–0.33% | 0.003–0.03% |

The level of risk is determined based on the level of risk acceptance and risk capacity. Different responses and measures are prescribed for different risks depending on their risk level, which are shown in Table 3. The levels of risk are categorized as 'unacceptable', 'marginally acceptable', 'acceptable', and 'ignorable', which are denoted by the indicators R1–R4, respectively. Table 4 details the different levels of risk. The risk level is determined based on the risk indicators compiled from the risk probability and risk impact. All possible combinations of risk probability and risk impact are presented in the risk matrix,

where users can find the risk level that corresponds to the risk probability and impact combinations.

**Table 3.** The risk matrix (modified from ITA, 2004).

| | | Risk Distribution | | | | |
|---|---|---|---|---|---|---|
| | | Risk Impact (G) | | | | |
| | | Minor G1 | Limited G2 | Severe G3 | Very Severe G4 | Catastrophic G5 |
| Risk probability (P) | Very Likely P5 | R2 | R1 | R1 | R1 | R1 |
| | Likely P4 | R3 | R2 | R2 | R1 | R1 |
| | Occasional P3 | R3 | R3 | R2 | R2 | R1 |
| | Unlikely P2 | R4 | R3 | R3 | R3 | R2 |
| | Very unlikely P1 | R4 | R4 | R4 | R4 | R3 |

**Table 4.** Risk levels (modified from ITA, 2004).

| Risk Level Standard | | |
|---|---|---|
| Risk Level | | Countermeasures |
| R1 | Unacceptable | Should not be included in project design, risk measures to be taken to reduce risk |
| R2 | Marginally acceptable | Risk mitigation measures to be taken |
| R3 | Acceptable | Include in the risk management process |
| R4 | Ignorable | Do not need to respond to this risk |

## 6. System Implementation

In this research, the risk management process was established for the MRT project, and IDEF0 was employed to analyze the operational process of each item. The MRT project is a large-scale project with a long construction duration; thus, it is very difficult and complicated to keep track of the risk management data at the different stages. Today, information systems are often used to manage large amounts of data, and databases can be designed in accordance with user requirements.

### 6.1. Database Design

By building a database system, all data can be controlled and managed together on a computerized platform, where the required information can be saved and accessed at the same time. In this way, the duplication and inconsistency of stored data can be significantly reduced, and data can be retrieved quickly. Most importantly, the format of the data is standardized. The advantages of a database system can only be realized through a detailed analysis and design to prevent compromising the integrity of the database.

### 6.2. Database Design Process

The first step in designing a database is to collect and analyze user requirements. The IDEF0 risk management process analysis is used to identify the inputs and outputs of each item. The users of the database are interviewed to determine their exact requirements and the data that must be stored in the database. The items that are not required are removed from the database, and the remaining items are arranged into a table for future use.

By collecting and analyzing requirements, a conceptual data model can be established for the general user. The model includes a simple description of the required user data, the relationship between data, data types, and so on. This will be developed into an entity–relationship model, which is implemented by the system designer. The model can be established by confirming the entity's property, verifying the relationships between entities, and establishing the basis of the entities' relationship. Then, a Business Database Management System (DBMS), such as Access and SQL Server, can be used to establish the database and set up the individual information spreadsheet, primary key and link, etc., to complete the design of the database.

*6.3. Entity-Relationship Model*

The Entity-Relationship Model (E-R Model) can be used to facilitate data analysis and design planning for the network and relational databases [13]. The aim of this research is to develop a risk management database system using the concept of the E-R model. The first step is to establish the required entity types, including the construction project, involved parties, work items, risk events, and disasters.

The next step is to characterize the relationships between the entities as one to one, one to many, or many to many. The four above-mentioned entities have affiliations within the project; for example, one plan can be divided into several sub-plans. For the different lines of the MRT project, the design can be divided into several sub-designs, the construction can be divided into several sub-construction sections, and the construction of each section can be divided into several sub-construction tenders. This means that the relationship is one to many. Construction projects contain several work items, and one work item contains several risk events, i.e., a one-to-many relationship. One risk event may cause several disasters, and one disaster can be caused by more than two risk events, which is a many-to-many relationship. Both the contractor and the supervisor are responsible for several risk work items, i.e., a many-to-many relationship.

The third step is to identify the properties of the entities and the relationships. The properties of a construction project include the project location, project background, project scope, etc. The properties of the involved parties include name, personnel, titles, and so on. The properties of a work item include the name of the work item, description, occurrence probability, influence level, and notes. The properties of a disaster include time, location, and response. An entity-relationship diagram with the properties of the entities is shown in Figure 4.

The users of the system are classified into five levels: system administrator, planning department, designer, construction contractor, and visitor. Users at different levels have different system modification and browse permissions. Each user has his or her own function items and website content level. The classification of the users is as follows.

First: System administrator:

The main jobs of the system administrator are to manage daily operations, system maintenance, and users' accounts (add or delete users) and to grant permissions to contractors for data that they are responsible for.

Second: Department/Contractor:

Normally, the Client is responsible for the project planning and is in charge of the preliminary stage of the project, along with the provision of basic project information and the risk policy. Designers, such as consultants, can identify and assess project risks after the system administrator assigns a new project to them. The contractor can review the risk assessment completed by the designer, develop responses to the risks, and establish a detailed risk response strategy for the construction stage.

Third: Visitor:

The settings for a visitor primarily allow the use of the system's search function. In order to protect the rights of the involved parties, visitors' accounts are usually created by the system administrator, and then they can browse the website and use the search function.

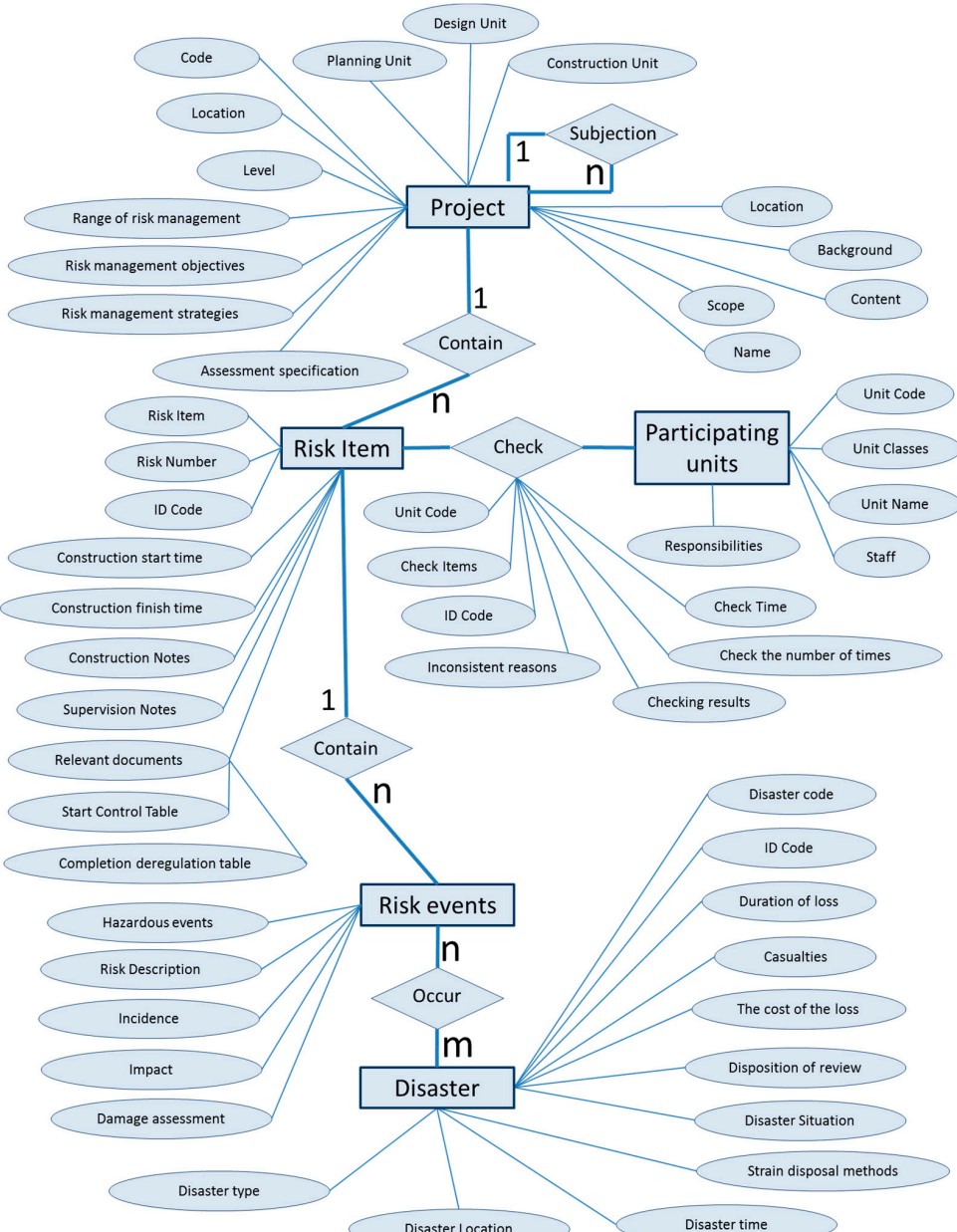

**Figure 4.** Entity-relationship diagram of the risk management database (including properties).

According to these principles, the plan of the website framework is shown in Figure 5.

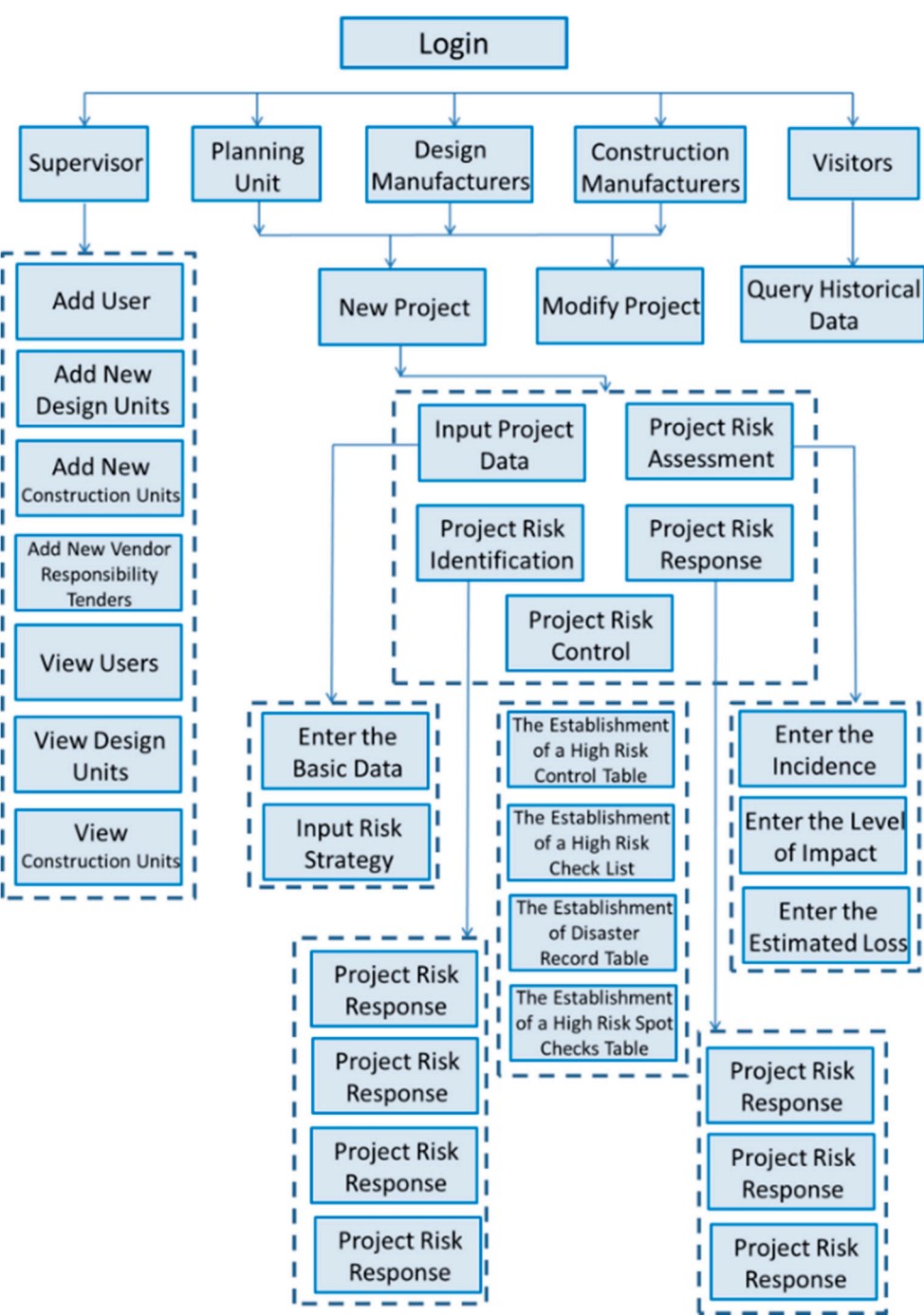

**Figure 5.** Framework of the website.

## 7. Case Study

This section collates the information required for establishing a database system for the risk management of public projects. First, the database is built; then, the needs of different database users are analyzed, and finally, the process is applied to a real project. The case study is the Taiwan Taoyuan International Airport Access MRT System Construction Project. This case was used to test the risk management process developed in this research.

### 7.1. Case Introduction

The Project of Taiwan Taoyuan International Airport Access MRT System is an MRT line that connects Taipei and Taoyuan International Airport. The total route length is

approximately 51.03 km. There are 22 stations, of which 15 are elevated and seven are underground, and two are maintenance depots. The Taipei Station (underground station), vehicle storage areas, and the building structure were constructed together. The viaduct section of the road segment contains a cut-and-cover tunnel section (about 1447 m long, including the excavated section and cut-and-cover section of about 586 m long) and the shield tunnel section (about 1584 m long).

*7.2. Risk Management in the Planning Stage of the MRT Project*

The first step is establishing basic data on the project, including the project name, the project type, the authority, the project location, the project background, the project scope, the project profiles, and other relevant information. The input flowchart of the planning stages of the program is shown in Figure 6.

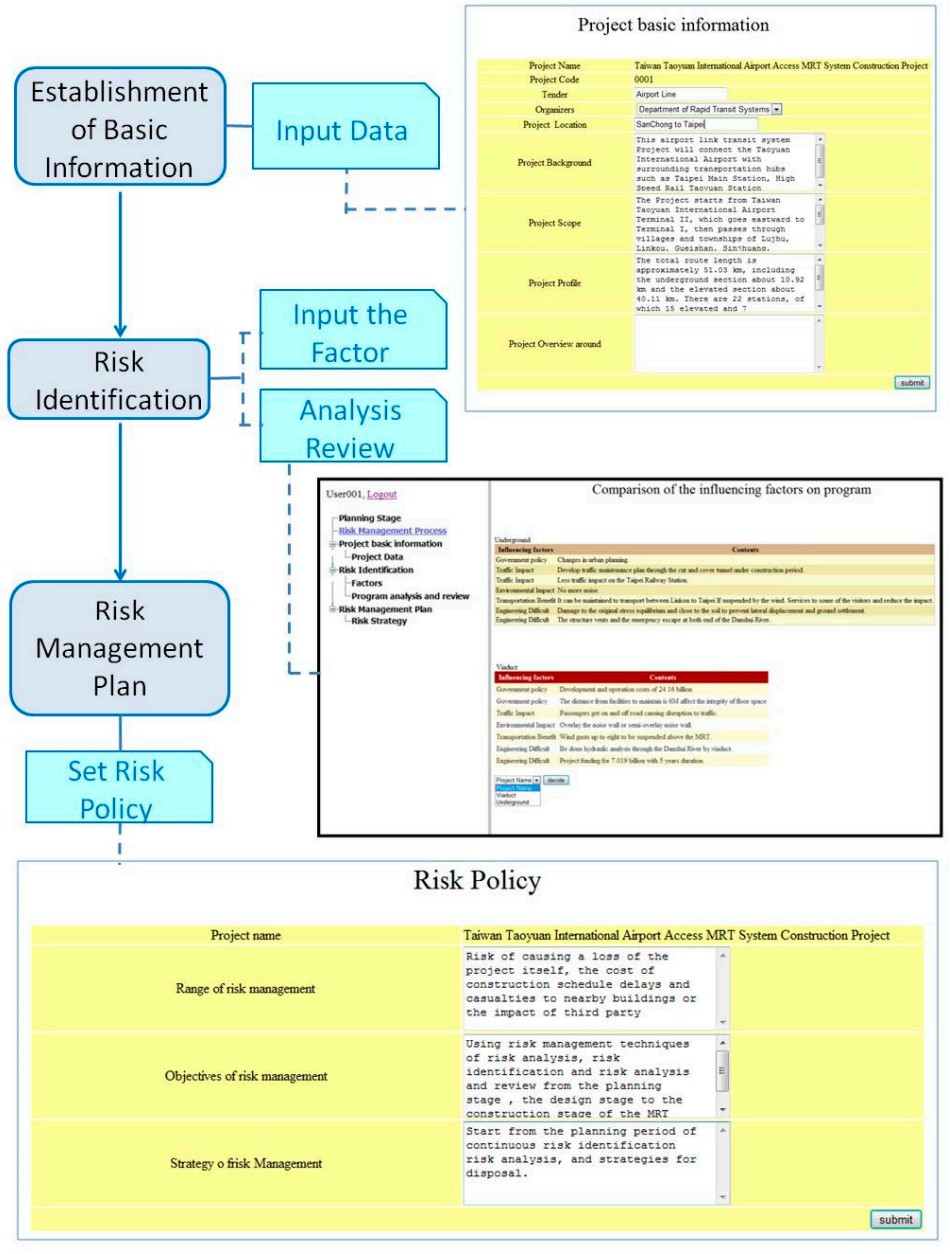

**Figure 6.** Planning stages of program input flowchart.

For the uncertainty assessment stage, two proposals resulted from different data collection and feasibility studies:

1.  The Railway Bureau proposed a viaduct approach.
2.  The MRT Taipei City Government Bureau proposed an underground approach.

The IDEF0 model was applied for risk management of the MRT Project. In this study, IDEF0 functional analysis and the MRT project risk management process was used to analyze the input, control, output, and mechanism of each stage of risk management. The analysis model is shown in Figures 7 and 8.

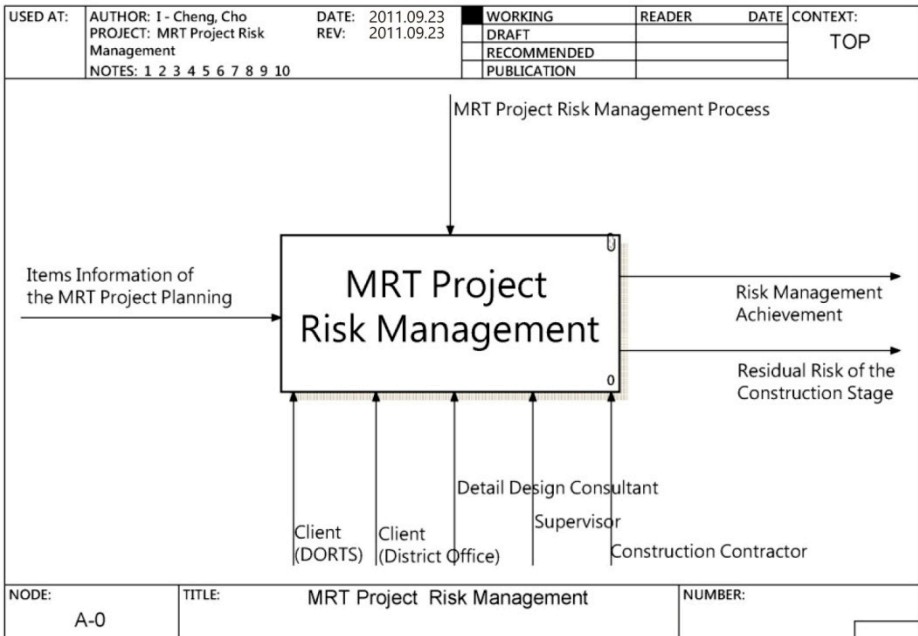

**Figure 7.** IDEF0 analysis model for MRT project risk management process.

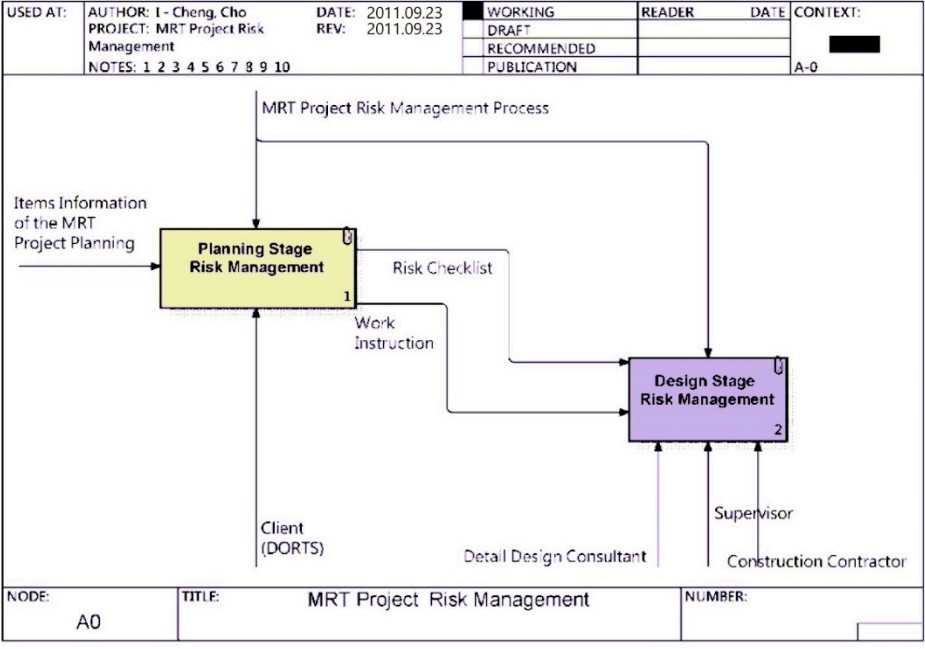

**Figure 8.** IDEF0 analysis model—A0 for the MRT project risk management process.

The model in this study is coded in accordance with the coding scheme of IDEF0. The first layer of the MRT project risk management is A0. The second layer includes A1

(risk management of the planning stage) and A2 (risk management of the design stage). Other stages are not shown in this paper. The third layer contains three items, from A11 to A13, which are risk assessment (A11), preliminary planning (A12), and determination of the requirements for the tender document for the detailed design (A13). The fourth layer comprises seven items (from A111 to A133), as shown in Table 5.

**Table 5.** IDEF0 coding principle.

| Item | First Level | Second Level | Third Level |
|---|---|---|---|
| A0 MRT Project Risk Management | A1 Planning Stage Risk Management | A11 Risk assessment | A111 Information collection |
| | | | A112 Influence factor identification |
| | | A12 Preliminary planning | A121 Line selection |
| | | | A122 Development of risk management plan |
| | | A13 Determination of requirements for the tender document for the detailed design | A131 Establishment of detailed design standard |
| | | | A132 Preparation of the detailed design tender document |
| | | | A133 Assessment selection on the detail design consultant |

A1: Planning stage risk management.

The first stage of the IDEF0 analysis model for the MRT project risk management process is the planning stage, code A1. The next stage is the third layer, including risk assessment, preliminary planning, and determination of the requirements for the tender documents for the detailed design. The fourth is the most detailed layer, which includes seven items. The input, control, output, and mechanism of the items in the fourth layer are described in Table 6 and Figure 9.

**Table 6.** IDEF0 analysis model—A1, planning stage.

| Node | No. | Stage Name | Input | Output | Control | Mechanism |
|---|---|---|---|---|---|---|
| A1 | 1 | Risk assessment | Information on items of the MRT project planning | Assessment report | MRT Project risk management process | Client |
| A1 | 2 | Preliminary planning | Assessment report | Risk checklist, risk policy, and assessment standard | MRT Project risk management process | Client |
| A1 | 3 | Determination of the requirements for the tender document for the detailed design | Risk policy and assessment standard | Work instruction | MRT Project risk management process | Client |
| A11 | 1 | Information collection | MRT project planning and all related information | Current land usage, urban planning information, and traffic volume demand | Complexity of the related information | Client |

**Table 6.** *Cont.*

| Node | No. | Stage Name | Input | Output | Control | Mechanism |
|------|-----|-----------|-------|--------|---------|-----------|
| A11 | 2 | Influence factor identification | Current land usage, urban planning information and traffic volume demand | Assessment report | Influence of construction duration and construction fund, third-party influences, surrounding environment | Client |
| A12 | 1 | Line selection | Assessment report | The line with the lowest risk level, risk checklist | Line configuration risk, station planning risk | Client |
| A12 | 2 | Development of risk management plan | The line with the lowest risk level | Work instruction | Establish risk policy and assessment standard, require the contractor establish the risk management plan | Client |
| A13 | 1 | Establishment of detailed design standard | Risk policy and assessment standard | Work scope, design standard | Divide the risk into general and special groups | Client |
| A13 | 2 | Preparation of the detailed design tender document | Work scope | Tender-related documents | Ensure the detailed design consultant has the technique to reduce the risk level and relevant risk management experience | Client |
| A13 | 3 | Assessment selection of the detailed design consultant | Tender-related documents | Selection methods | Ensure the detail design consultant has the technique to reduce the risk level and relevant risk management experience | Client |

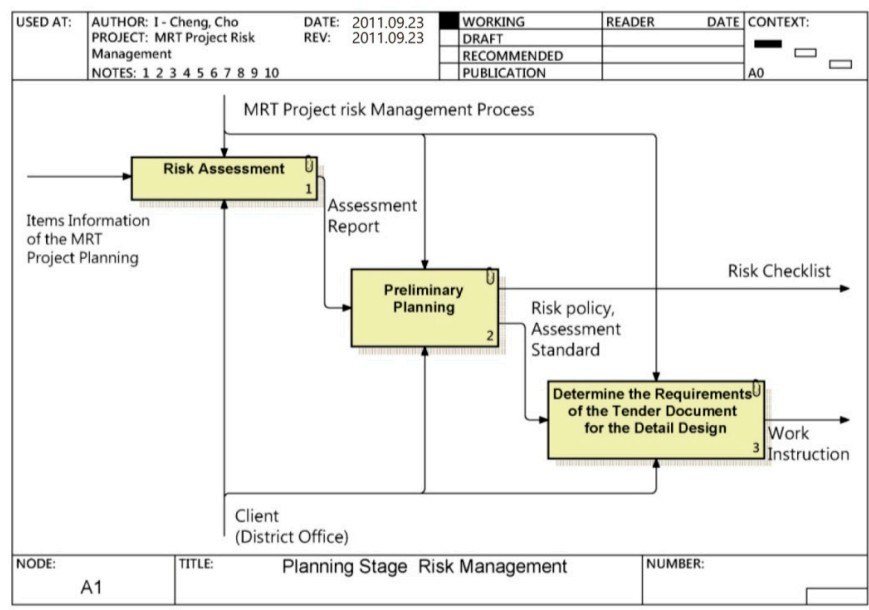

**Figure 9.** IDEF0 analysis model—A1, for the MRT project risk management process.

The input, control, output, and mechanism of the items in the IDEF0 analysis model for the planning stage of risk management are described in detail below. The IDEF0 analyses for A11, A12, and A13 are illustrated in Figures 10–12, respectively.

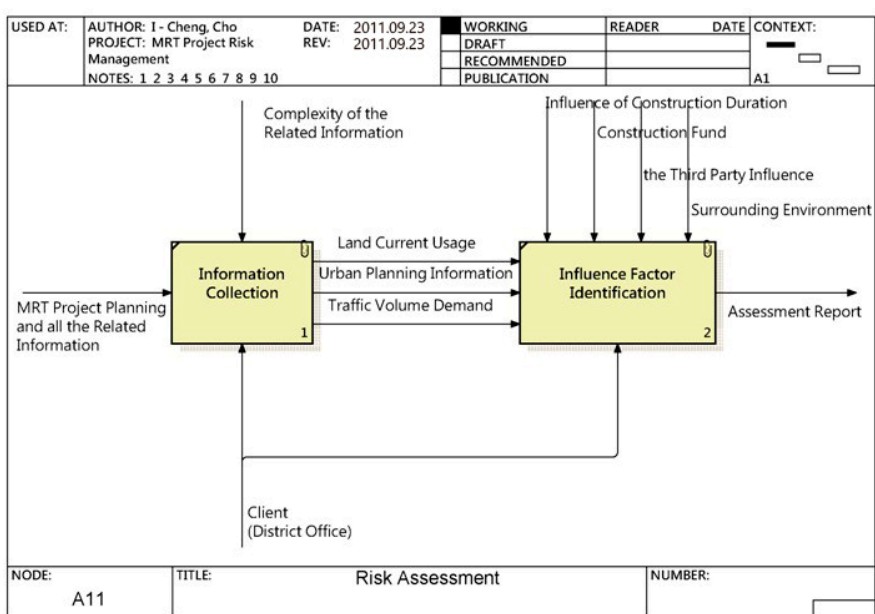

**Figure 10.** IDEF0 analysis model—A11, for the MRT project risk management process.

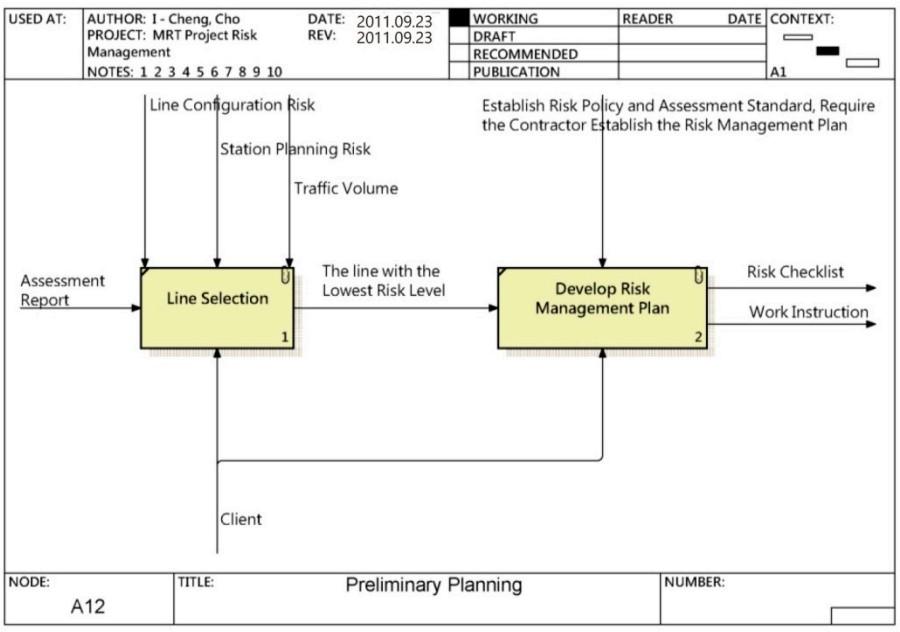

**Figure 11.** IDEF0 analysis model—A12, for the MRT project risk management process.

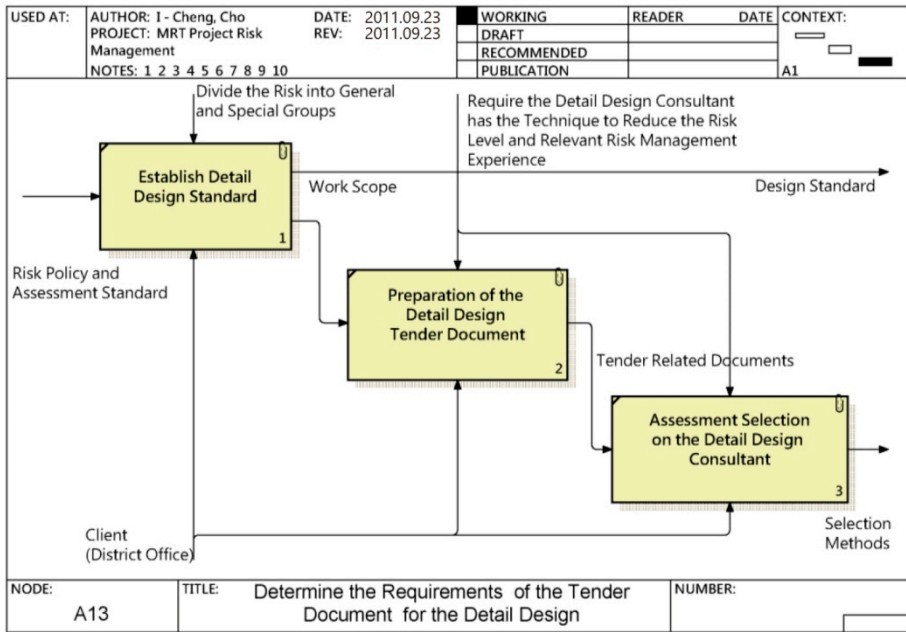

**Figure 12.** IDEF0 analysis model—A13, for the MRT project risk management process.

According to the above analysis, the viaduct proposal has the combined advantages of lower costs, shorter duration, and fewer construction difficulties. On the other hand, the underground proposal, while more expensive, is the better choice when considering long-term urban plan development, the safety and convenience of passengers, management, land acquisition and development, environmental impact, and net benefits. The benefits of the underground proposal compensate for the higher construction costs. Thus, the underground proposal was chosen for future development.

After the preliminary design in the planning stage, the high-risk items are transferred to the next stage of the project. Thus, the detailed design tender documents must clearly describe the high-risk items and state the skills and experience required of the contractor/consultant. For example, in this project, the open-cut construction method and the shield method (TBM) should be stated in the tender documents by the owner.

The consultants should identify possible risks during the detailed design phase and record them as risk items, as shown in Table 7. The next step is numbering the identified risk items using the reference coding scheme, as illustrated in Figure 13. The input flowchart of the design stages of the program is shown in Figure 6.

**Table 7.** Risk items.

| Preliminary Risk Identification: Detailed List | | | |
|---|---|---|---|
| **Project** | **DA115** | **Report Unit** | **Risk Management Team** |
| **No.** | **Risk Event** | | |
| 1 | The investigation has inadequate funding and a tight schedule | | |
| 2 | Stakeholder requirements for content is not clear | | |
| 3 | Requirements for the implementation of the phase process is not clear | | |
| 4 | Design quality and schedule management | | |

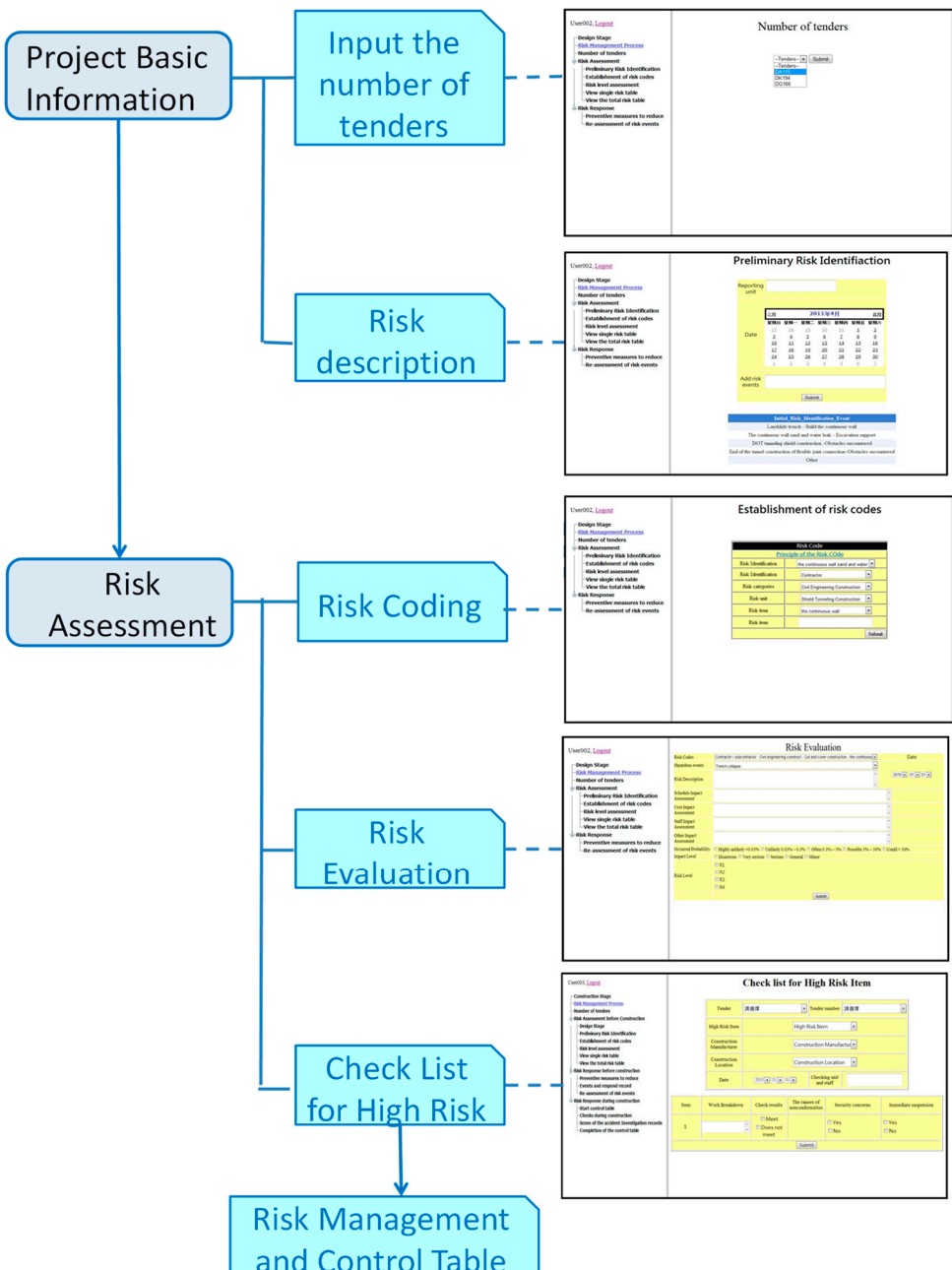

**Figure 13.** Design stages of program input flowchart.

After encoding possible hazards of the identified risk items, the next step is to perform a risk analysis and risk assessment. The risk analysis provides a detailed description of the item from four aspects: schedule, cost, personal injury, and other influences. This is necessary to enable the careful evaluation and recognition of risk. The risk assessment involves expert grading methods that transform the identified project risks into a questionnaire, which is used by a grader panel to assess the occurrence probability, consequences, and level of each risk. Evaluators also provide relevant comments as a reference for risk elimination measures.

The grader panel is composed of a risk assessment team, which is established by the design unit, and external professional consultants (seven people in total with more than 10 years of professional experience). Based on the assessment specifications established for this project, the panel evaluates each risk item and determines the occurrence probability and the impact level of the risk.

After the proposal of risk prevention or reduction measures for hazards or events caused by risk items with high initial risk levels, the managed risk level of each item is evaluated. These outcomes are updated in a risk management and control table. A part of the risk management and control table that is transferred after the completion of the planning stage to the design phase is illustrated in Table 8.

In the risk management and control table (after the risk item has been managed through countermeasures), the decrease or increase in the risk level of each item and the difference in the risk level before and after countermeasures will appear in a risk profile diagram. For example, the risk item with a series number of 01 is "insufficient funding for survey, tight schedule"; the original occurrence probability is P5, the original impact is G4, and the original risk level is R1. After the risk is managed with countermeasures and re-evaluated, the occurrence probability of the managed risk item becomes P2, its impact decreases to G1, and its risk level is reduced to R4. Changes in the risk level can be fully displayed in the risk profile diagram, which is shown in Figure 14.

**Table 8.** Risk management and control table.

| No. | Risk Item | Original Risk Probability | Original Risk Impact | Original Risk Levels | Final Risk Probability | Final Risk Impact | Final Risk Levels |
|-----|-----------|--------------------------|---------------------|---------------------|------------------------|-------------------|-------------------|
| 1 | Insufficient funding for surveying and tight schedule | P5 | G4 | R1 | P2 | G1 | R4 |
| 2 | Unclear stakeholder requirements | P4 | G4 | R1 | P1 | G1 | R4 |
| 3 | Unclear scheduling requirements | P4 | G4 | R1 | P2 | G1 | R4 |
| 4 | Poor design quality and progress management | P4 | G3 | R2 | P1 | G1 | R4 |

**Figure 14.** Risk matrix.

The managed residual risk is further transferred to the next stage and prevents hazards caused by construction risk. The detailed design consultants must summarize the risk management outcomes and convey them to construction companies. They must provide sufficient information to the construction company and draft a complete construction specification based on the design outcome for general and special projects to assist the construction manufacturers.

To consolidate the Metropolitan Rail Transit project from the planning stage to the design stage, the risk items identified in the initial stage are ranked and arranged from

highest to lowest risk. The control and appropriate management of the risk items are tracked accordingly. At the detailed design stage, the listed risk items are successively checked to determine whether the design can reduce or transfer the risks of the item listed in the risk management table. Risk items are deferred to the subsequent construction phase when they cannot be reduced or transferred in the design stage.

## 8. Conclusions

Public construction projects are characterized by complexity, long durations, and a large impact on society. Thus, the success of a public construction greatly relies on proper risk management. In each phase of the project, the sources, impacts, and responses to risks should be studied and managed properly so that with effective tracking and control, the hazardous results of risks can be eliminated or minimized.

The steps of this research are as follows. Firstly, the literature related to risk management was thoroughly studied to understand the underlying theory and process. Secondly, a risk management model was established by combining the risk management method of the Project Management Institute (PMI) and ITA. The Taiwan Taoyuan International Airport Access MRT System Construction Project was used as a real case study to implement and further modify the process of risk management for public construction projects. In the next stage, IDEF0 was used to analyze the implementation of the risk management model utilizing the syntax of input, control, output, and function terms, as well as the various roles played by owner.

This study analyzed the detailed mechanism and procedure of the information flow between the design consultant, the supervision unit, and the construction manufacturer. The information was collected in a table and used to build a database. Finally, as an example, the risk management process and the database constructed in this study were applied to the Taoyuan Airport MRT project. The real case study demonstrates that the proposed approach can indeed achieve efficient risk management and reduce project risks. The number of risk management projects dealt with every year of the seven-year project execution was gradually reduced until the project was closed. The results of this research can be used as a reference for the risk management process of public buildings in the Figure 15.

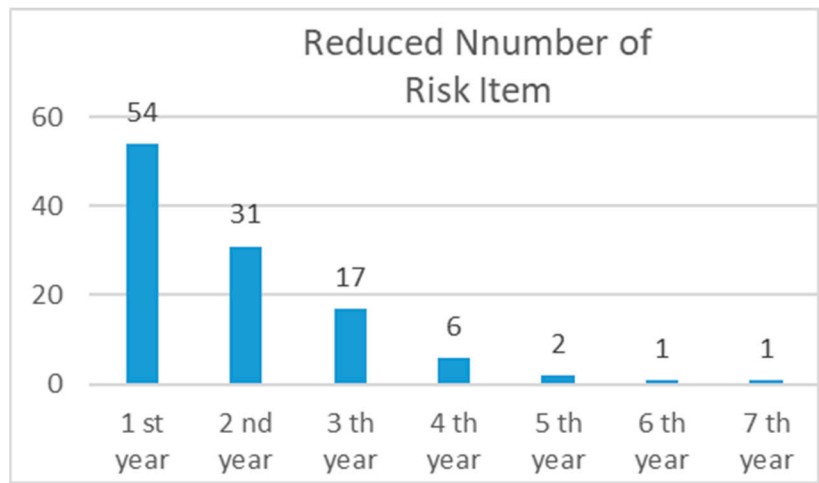

**Figure 15.** Reduced number of risk item.

This study aimed to establish and develop a risk management process for public construction projects by using foreign norms for the assessment component of the ITA. However, the same set of assessment models cannot be applied to all engineering projects, which will have different project properties/characteristics. Therefore, it is recommended

that future research projects explore different forms of projects to establish different standards and improve the accuracy of evaluation.

**Author Contributions:** Y.-F.L. provided the related literature review and part of the flowchart and classification table of IDEF0; C.-H.C. provided the information and content related to MRT construction; H.-P.T. provided confirmation of the research direction of the article; I.-C.C. completed the writing of the manuscript, process integration and case analysis content and results integration of actual implementation cases. All authors have read and agreed to the published version of the manuscript.

**Funding:** This research received no external funding.

**Institutional Review Board Statement:** Not applicable.

**Informed Consent Statement:** Not applicable.

**Data Availability Statement:** Data sharing not applicable. No new data were created or analyzed in this study. Data sharing is not applicable to this article.

**Conflicts of Interest:** The authors declare no conflict of interest.

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
