# Peer review of "Developing a Risk Management Process for Infrastructure Projects Using IDEF0"

_sustainability, doi:10.3390/su13126958_

Round 1

Reviewer 2 Report

Of interest and important. Only three detailed remarks:

1.the explanation of acronym IDEFO appears in the line 128. That is rather late. It should at least in the Abstract .Perhaps even in the title .

2. The range of keywords (4x) sounds very  modest contrary to the rich content of the paper.

3.The last line of conclusions "  in order to assess the results more accurate"        seems to be too obvious.  

Round 2

Reviewer 1 Report

Thank you for your effort in revising the manuscript. However, I can see that most of the given comments have been ignored. Therefore, I see no point in the publication of this manuscript at its current form. 

Best Regards,

Referee

Author Response

Please see the attachment, thank you

Round 3

Reviewer 1 Report

In my point of view, this paper is not ready for the publication due to its inappropriate structure, as I have explained in my previous review reports. Having top quaility in presenting a conducted research (writing and structuring the paper) is even more important than the quality of research itself and its findings. I would like to see relevant and specific actions concerning my comments, not just some general responses. Please carefully look at my comments in my first review report and take the required actions accordingly.

Author Response

This manuscript is a resubmission of an earlier submission. The following is a list of the peer review reports and author responses from that submission.